# The Structure and Magnetic Properties of Rapidly Quenched Fe_72_Ni_8_Nb_4_Si_2_B_14_ Alloy

**DOI:** 10.3390/ma14010005

**Published:** 2020-12-22

**Authors:** Lukasz Hawelek, Tymon Warski, Patryk Wlodarczyk, Marcin Polak, Przemyslaw Zackiewicz, Wojciech Maziarz, Anna Wojcik, Magdalena Steczkowska-Kempka, Aleksandra Kolano-Burian

**Affiliations:** 1Lukasiewicz Research Network—Institute of Non-Ferrous Metals, 5 Sowinskiego str., 44-100 Gliwice, Poland; tymon.warski@imn.gliwice.pl (T.W.); patryk.wlodarczyk@imn.gliwice.pl (P.W.); marcin.polak@imn.gliwice.pl (M.P.); przemyslaw.zackiewicz@imn.gliwice.pl (P.Z.); magdalena.steczkowska-kempka@imn.gliwice.pl (M.S.-K.); olak@imn.gliwice.pl (A.K.-B.); 2Institute of Metallurgy and Materials Science, Polish Academy of Sciences, 25 Reymonta str., 30-059 Krakow, Poland; w.maziarz@imim.pl (W.M.); wojcik.a@imim.pl (A.W.)

**Keywords:** soft magnetic materials, metallic glass, crystallization, magnetic properties

## Abstract

The complex structural and magnetic studies of the annealed rapidly quenched Cu-free Fe_72_Ni_8_Nb_4_Si_2_B_14_ alloy (metallic ribbons form) are reported here. Based on the calorimetric results, the conventional heat treatment process (with heating rate 10 °C/min and subsequent isothermal annealing for 20 min) for wound toroidal cores has been optimized to obtain the least lossy magnetic properties (for the minimum value of coercivity and magnetic core losses at 50 Hz). For optimal conditions, the complex permeability in the 10^4^–10^8^ Hz frequency range together with core power losses obtained from magnetic induction dependence up to the frequency of 400 kHz was successfully measured. The average and local crystal structure was investigated by the use of the X-ray diffraction method and the transmission electron microscopy observations and proved its fully glassy state. Additionally, for the three temperature values, i.e., 310, 340 and 370 °C, the glass relaxation process study in the function of annealing time was carried out to obtain a deeper insight into the soft magnetic properties: magnetic permeability and cut-off frequency. For this type of Cu-free soft magnetic materials, the control of glass relaxation process (time and temperature) is extremely important to obtain proper magnetic properties.

## 1. Introduction

Soft magnetic materials (SMMs) are still of great interest in motor [1], power converter [2,3], switched-mode power supplies [4] and sensor applications [5]. Currently, amorphous and nanocrystalline Fe-, Co- and Ni-based alloys (or combination of ferromagnetic metals) can be used for this purpose [6,7,8,9]. There are two main approaches in developing and optimizing of novel SMMs. The first one is aimed at maximizing saturation induction (B_s_), while the second one in minimizing coercivity (H_c_) and core losses (P_s_). The FeCo- and Co-based alloys are reported to have very high induction saturation, good thermal stability and improved mechanical properties. Unfortunately, Co is relatively expensive and is classified as a critical material [10]. It was also recently found that only alloys with low Co content can have good soft-magnetic properties such as low H_c_ and high effective permeability (µ_e_) [11,12]. FeNi-based SMMs are a newer class of materials for perspective application at higher frequencies for motors and converters, in which the most important advantage is extremely low coercivity and core losses value, as it was shown in the previous studies [13,14]. In these studies, the structural and magnetic properties of the Cu-free (Fe_100−x_Ni_x_)_80_Nb_4_Si_2_B_14_ alloys were investigated. The magnetization measurements performed on heat-treated samples shown that B_s_ value is decreasing almost linearly from 1.6 T for Ni = 10 at.% to 0.75 T for Ni = 70 at.% content. The Curie temperature of as-cast ribbons is increasing from 320 °C for Ni = 10 at.% up to 400 °C for Ni = 30 at.% and then it is decreasing for higher Ni content alloys. The Curie point was found to be lower than the temperature of primary crystallization peak observed by calorimetry. However, K. Suzuki et al. in 2001 [15] showed that nanocrystallization of the Cu-free Fe-Nb-B alloys is most likely caused by a high homogenous nucleation rate in the supercooled liquid regime as well as a delayed growth rate induced by a large redistribution of Nb. Additionally, the most present study shows that for Cu-free alloys the lowest value of H_c_ exists before the nanocrystallization in the so called “stress relief” stage [16]. Therefore, the detailed studies of crystalline structure and magnetic properties evolution in the function of annealing temperature (T_a_) for Cu-free FeNi-based alloys are of great importance.

In the present work, the complex structural and magnetic study of Fe_72_Ni_8_Nb_4_Si_2_B_14_ amorphous alloy are presented. Thermal stability, structure and nanostructure evolution of annealed samples are examined by the use of differential scanning calorimetry (DSC), X-ray diffraction (XRD) and transmission electron microscopy (TEM) observations. Magnetic parameters were obtained by measurements of B(H) hysteresis loops, H_c_, B_s_, core losses (P_s_), complex permeability µ and P_s_ in the frequency range (50 Hz–400 kHz). Additionally, the controlled aging process at three different temperatures has been performed to verify the magnetic permeability and cut-off frequencies as a function of annealing time.

## 2. Materials and Methods

Precursors for amorphous Fe_72_Ni_8_Nb_4_Si_2_B_14_ alloy were prepared from pure chemical elements Fe (3N), Ni (3N), Si (4N) and the binary compounds FeB_18_ (2.5N), FeNb_65_ (2.5N) using an induction furnace in an argon atmosphere (heating at 1450 °C for 20 min, casting at 1250 °C). The amorphous alloy in the form of ribbons 28 µm thick and 6.5 mm wide by the melt spinning technique (at 30 m/s Cu wheel speed and casting temperature at 1250 °C). To achieve the optimal magnetic parameters (min. value of H_c_ and P_s_ at 50 Hz and B = 1T), the toroidal cores (inner and outer diameter of 20 and 30 mm, respectively) were isothermally annealed for 20 min in a vacuum furnace (5·10^−4^ mbar) at different temperatures (from 340 to 440 °C). Additionally, the effect of aging at three different temperatures: 310, 340 and 370 °C (far below the crystallization temperature) for up to 6200 min has been checked and the evolution of magnetic permeability and cut-off frequency have been monitored. The amorphousness of the as-spun and annealed ribbons were studied by X-ray diffraction (XRD) at room temperature using Rigaku MiniFlex 600 diffractometer (Rigaku, Tokyo, Japan) equipped with copper tube CuKα. The crystallization processes have been monitored by the differential scanning calorimetry (DSC) with a heating rate of 5 –30°C/min using thermal analyzer Netzsch 404C Pegasus (NETZSCH-Gerätebau GmbH, Selb, Germany). The transmission electron microscopy (TEM) images in the bright-field (BF) mode and selected area diffraction patterns (SADPs) were recorded using Tecnai G2 F20 (200kV) electron microscope (Thermo Fisher Scientific, Waltham, MA, USA). Thin foils for TEM observations were prepared with TenuPol-5 double jet electropolisher using an electrolyte of perchloric acid (80%) and methanol (20%) at temperature near −20 °C. The Remacomp C-1200 (MAGNET-PHYSIK Dr. Steingroever GmbH, Köln, Germany) magnetic measurement system was used to determine B(H) and P_s_. The complex magnetic permeability in the frequency (f) range 10^4^–10^8^ Hz at room temperature of the toroidal cores was measured using impedance analyzer Agilent 4294A (Agilent, Santa Clara, CA, USA).

## 3. Result and Discussion

For melt-spun ribbon, the α-Fe type phase crystallization kinetics has been studied by use of the differential scanning calorimetry (DSC) by performing measurements with heating rates in the range from 5 to 30 °C/min. Obtained DSC curves have been plotted in Figure 1a. The onset temperature of crystallization peak varies from 479.7 °C for heating rate 5 °C/min up to 500.7 °C for heating rate 30 °C/min. For such non-isothermal crystallization processes, the Kissinger model [17] was used to determine the average activation energy. This method is based on the equation:(1)$$ln\left(\frac{\phi }{T_{p}^{2}}\right)=ln\left(\frac{A_{0}R}{E_{a}}\right)-\frac{E_{a}}{\left(RT_{p}\right)},$$
where ϕ is a heating rate, $T_{p}$—the temperature of the crystallization peak, $E_{a}$—activation energy, R—gas constant and $A_{0}$—pre-exponential factor. By linear fitting of $\ln\left(\frac{\phi }{T_{p}^{2}}\right)$ vs. $\frac{1}{T_{p}}$ curve the average activation energy E_a_ of the process has been determined from the slope of this curve (Figure 1b). The obtained average activation energy E_a_ of crystallization of the α-Fe is equal to 430.2 ± 6.5 kJ/mol. Crystallization peaks were also fitted to the Avrami equation:(2)ln(−ln [1 −α(t)])=ln(k)+nln(t)
where: α(t)—degree of crystallization, k—crystallization rate constant, n—Avrami constant, t—time. All the Avrami fits were conducted in the 30–70% range of the degree of crystallization and gathered in Figure 1b. From the Avrami fits, average n index describing the mechanism of crystallization was derived. It is equal to 2.5 ± 0.2. According to Malek [18], the kinetic exponent n = 2.5 describes the transformation as the diffusion controlled growth process with the constant nucleation rate.

In Figure 2, the dependences of the coercivity H_c_ on the annealing temperature T_a_ with the magnetic saturation B_s_ (T_a_) Figure 2a and on the core losses P_s_ (T_a_) Figure 2b are presented. H_c_ and B_s_ values were taken from the hysteresis loops measured up to magnetic saturation state. The H_c_ (T_a_) curve shows the minimum value at T_a_ = 370 °C with H_c_ = 3.95 A/m and B_s_ = 1.09 T. Below this temperature H_c_ value is decreasing from 5 A/m at 340 °C, while above this temperature H_c_ value is increasing up to 125 A/m at 440 °C. The magnetic saturation is increasing with T_a_ from 1.05T at 340 °C up to 1.29 T at 440 °C. The Bs value at coercivity minimum equals 1.09 T. Obtained B_s_ is much lower than presented in [3] for Fe_70_Ni_10_Nb_4_Si_2_B_14_, but, as was mentioned in the introduction section, authors did not optimize the annealing temperature and didn’t measure H_c_ and P_s_ values. From an application point of view the knowledge of H_c_ and especially of P_s_ is of the greatest importance. It is clearly seen in Figure 2, that P_s_ (T_a_) dependence shape strongly correlates with the H_c_ one. The minimum value P_s_ = 0.092 W/kg coexists with a minimum value of H_c_ at annealing temperature T_a_ = 370 °C. For sample annealed in P_s_ minimum (so-called optimal conditions) at 370 °C the complex magnetic permeability has been measured at room temperature (RT) in order to obtain the level of real µ’ and the frequency value of imaginary µ” maximum. Both components are gathered in Figure 3. The magnetic permeability µ’ reaches 3100 in low frequency limit (f = 10^4^ Hz) and decreases for higher frequencies, while the maximum value of magnetic permeability loss µ” exists for f = 5·10^5^ Hz. This value is crucial for application purposes and is defined as “cut-off” frequency, the usage frequency of this material [19,20]. The next application important value of soft magnetic materials are the AC core power losses P_s_ in the function of B_s_ for the low (50 Hz) to high (400 kHz) frequencies. This log–log dependence is shown in Figure 3. It can be seen that, for all the frequencies, the P_s_ value increases almost linearly with increasing magnetic field strength, mainly because more immense energy is required to increase induction near the saturation [4]. Obtained magnetic properties have been finally collected and compared with the other previously reported FeNi-based soft-magnetic alloys in Table 1. As shown, there are still some gaps in the data on the magnetic properties of classically annealed FeNi-based alloys. Moreover, when using a rapid annealing process, very attractive magnetic properties must be obtained.

As was shown by Yoshizawa [22], small additions of Cu and Nb facilitate the formation of an amorphous-crystalline nanocomposite microstructure upon annealing of initially amorphous precursor materials. For the Cu-free and Nb-containing alloy, as we study in this work—Fe_72_Ni_8_Nb_4_Si_2_B_14_—the Nb atoms play commonly a diffuse role towards the surrounding amorphous matrix and increase the thermal stability of the matrix. The XRD patterns of annealed samples gathered in Figure 4 prove that, for temperature annealing up to 400 °C for 20 min, the crystal structure remains amorphous and only first and second-order diffused amorphous halos exist on the patterns. For annealed sample at T_a_ = 420 °C, the three small diffraction peaks are visible and correspond to the α-Fe phase, while for T_a_ = 440 °C these peaks are of higher intensity. The TEM observations in BF mode (Figure 5a,c) and SADPs (Figure 5b,d) for samples annealed at 370 and 420 °C, respectively, proved the amorphous state of the annealed sample at 370 °C and presence of ~10–30 nm α-Fe nanocrystals in the second one. Based on the TEM observations, it can be seen that, for a sample with optimal magnetic properties (e.g., the lowest P_s_), the crystal structure in the nano scale remains in the amorphous state.

Aging of metallic glasses is caused by the annealing process. Structural relaxation linked with quenched-in stress elimination is associated with the improvement of the magnetic properties of the soft magnetic amorphous ribbon. From the point of view of the crystal structure, aging in metastable quenched metallic glasses induces a lower enthalpy, a smaller volume, a more stable glassy state and changes the topological short-range order, which is characteristic for the glass structure [23,24,25,26,27,28] Some of the previous studies have shown that the relaxation process of FeNiSiB systems can be divided into two stages: the first—metalloid atoms movement, the second—diffusion of the constituent atoms [29]. From the magnetic point of view, the elimination of internal stresses can improve the mobility of the Bloch wall of the magnetic domain [30] and, as a consequence, the magnetic anisotropy fluctuates during the change of the topological short-range order caused by the aging/rejuvenation process [31]. As the relative permeability is inversely proportional to the anisotropy constant, the magnetic saturation increases with the local structure change and the decrease in anisotropy constant [32]. Thus, based on the above and on the fact that optimal annealing conditions (in the context of minimum H_c_ and P_s_ value) are treatment at 370 °C for 20 min and the material is still in the glassy state, further verification of the annealing process in the glassy state has been performed also at lower temperatures: 310, 340 and 370 °C, with different annealing times up to 6200 min. As has been shown in Figure 6 the µ’ value increases substantially from 3100 obtained for 20 min of annealing up to over 4500 for annealing time in the range 200–2000 min. Interestingly, for lower annealing temperature T_a_ = 340 °C µ’ consequently increases with annealing time and reaches 5000 for 6200 min. Process of glass relaxation is much weaker for Ta = 310 °C and for the annealing time up to 6200 min µ’ slightly increases. It has been noted that the relaxation process is still not complete. The cut-off frequency for Ta dependence indicates how the annealing process affects the µ” peak position that may be correlated with the reorganization of the magnetic domains during glass relaxation [30]. The downward trend in the cut-off frequency suggest that slow reorganization of the local structure occurs during aging process and some α-Fe clusters may form and grow. As was shown here, from the energy point of view, for lower temperature (340 °C) i.e., when less energy is delivered to the glass system, a slower relaxation process takes place and a higher permeability value in relaxed glass can be achieved. Considering the stability of cut-off frequency parameter, the most stable (invariant) annealing temperature is 370 °C and there are no significant changes in the annealing time up to 6200 min. This is crucial from the application point of view and needs to be investigated more closely soon. For all the samples after such aging process the XRD measurements proved the amorphous state of the samples (see Figure 7). As it was shown in some recent papers [33,34,35,36], the APT (Atom Probe Tomography) together with MOKE (Magneto-Optic Kerr Effect) and in situ AFM (Atomic Force Microscopy) observations should give us a more precise explanation of the glass relaxation process and early stage of the nucleation process in the studied system. The APT should give us the information on Fe, Ni as well as B, Si partitioning in the glassy state during long-term aging and moment of initial state of crystallization, while MOKE&AFM will allow to monitor the surface magnetic properties and morphology changes induced by relaxation process.

## 4. Conclusions

The structural and magnetic properties of the conventionally annealed Fe_72_Ni_8_Nb_4_Si_2_B_14_ alloy prepared by melt spinning have been investigated. The deep insight into the correlation between the magnetic properties B_s_, H_c_, P_s_ and T_a_ allowed to determine the optimal conditions. For the T_a_ = 370 °C, minimum value of H_c_ of 3.95 A/m, P_10/50_ of 0.092 W/kg, B_s_ = 1.09T, µ’ of 3100 and cut-off frequency of 5 × 10^5^ Hz have been obtained. That is over 100 °C below the crystallization onset temperature obtained from the DSC study. The P_s_ (B) dependence has been also shown and the optimal annealed material possess the linear P_s_ (B) dependence up to 400 kHz. The structural study has shown that, for such optimal conditions, material remains in the glassy state; the so-called relaxed glassy state. Deeper insight into the relaxation process occurring during the aging of this metallic glass shown that is possible to get higher µ’ values for longer annealing (over 100 h) time even at lower temperature T_a_ = 340 °C. However, the optimal T_a_ = 370 °C is the most stable from the point of view of the stability of cut-off frequency value.

## Figures and Tables

**Figure 1 materials-14-00005-f001:**
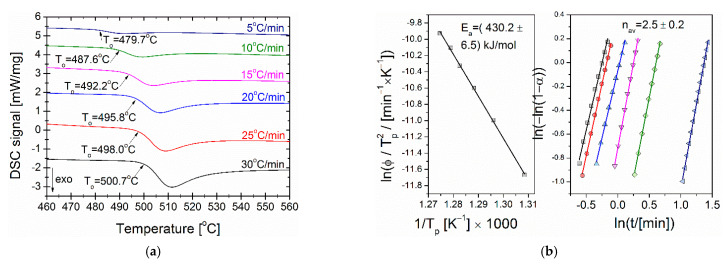
DSC signals (**a**), activation energy E_a_ of α-Fe phase crystallization and the Avrami exponent n (**b**) estimated for as-spun metallic glass (colors defines heating rates as shown in Figure 1a).

**Figure 2 materials-14-00005-f002:**
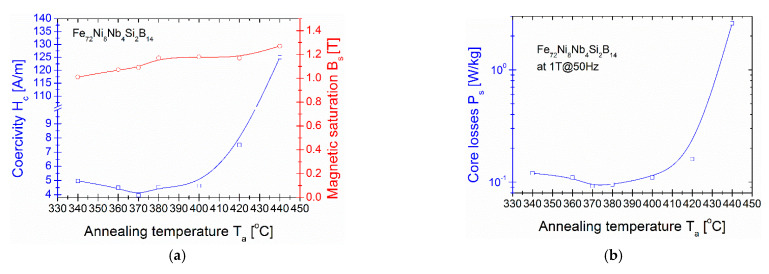
H_c_ and B_s_ from T_a_(20mins) (**a**) and P_s_ from T_a_ (**b**) dependences. The lines are only guide for eyes and not a fit.

**Figure 3 materials-14-00005-f003:**
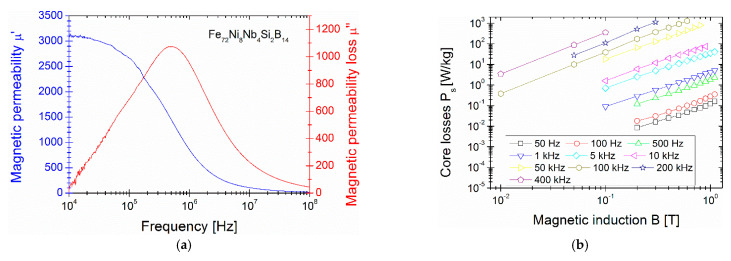
Magnetic permeability µ’ (**a**) and magnetic permeability loss µ” (**b**) dependence in the function of frequency 10^4^–10^8^ Hz for sample annealed at 370 °C.

**Figure 4 materials-14-00005-f004:**
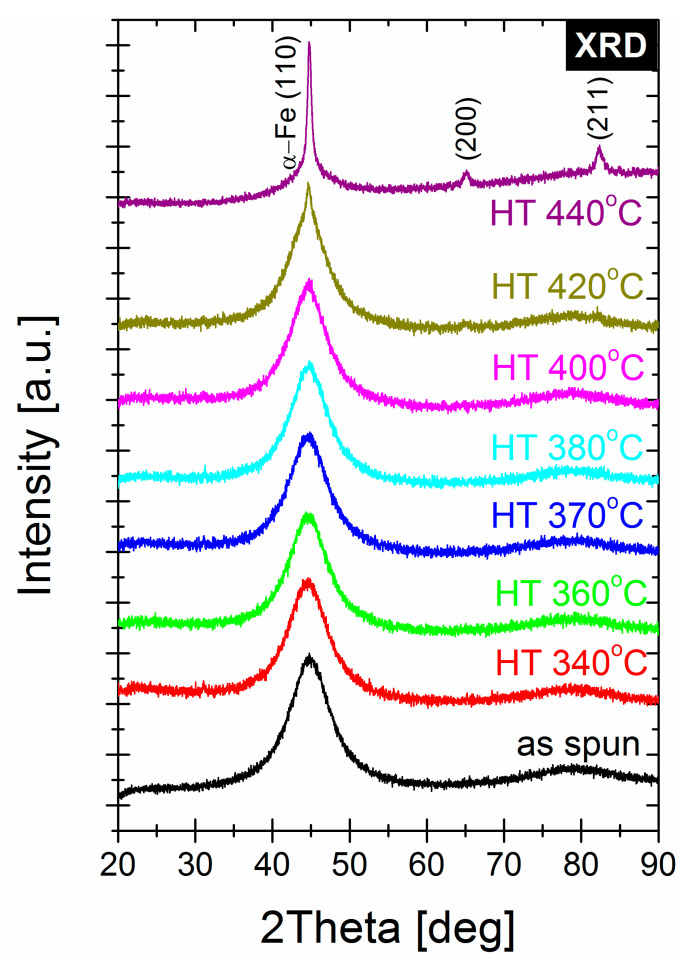
XRD patterns for annealed samples.

**Figure 5 materials-14-00005-f005:**
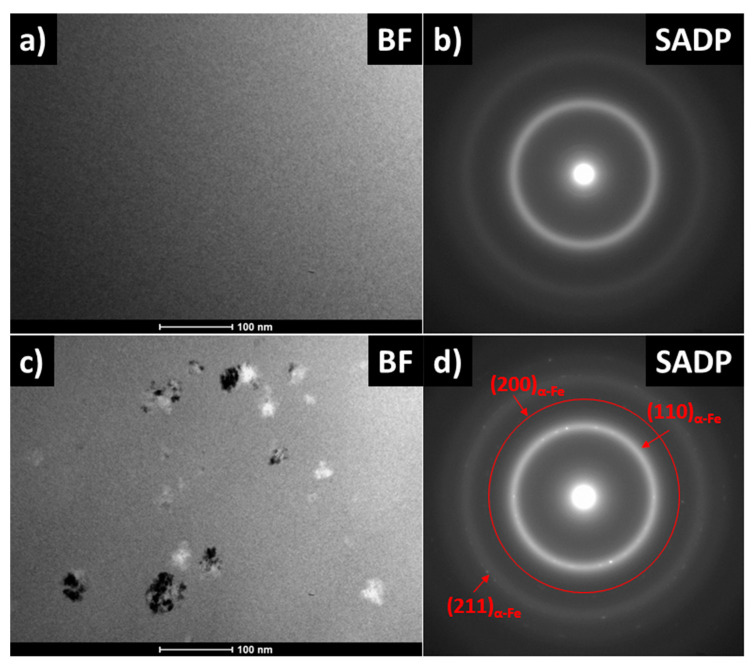
TEM images for annealed samples: (**a**) BF at 370 °C (**b**) SADP at 370 °C (**c**) BF at 420 °C (**d**) Scheme 420 °C.

**Figure 6 materials-14-00005-f006:**
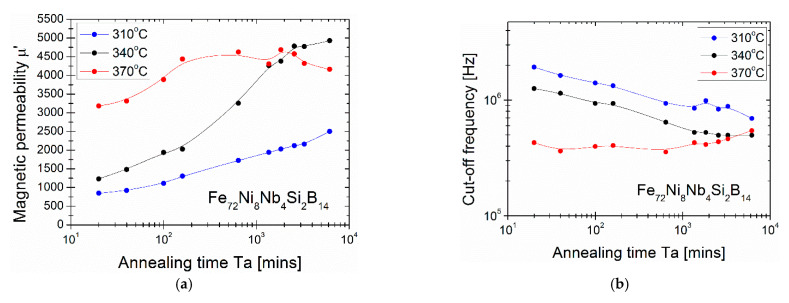
Magnetic permeability (**a**) and cut-off frequency (**b**) dependencies on annealing time performed at 310 °C, 340 °C and 370 °C. The lines are only guide for eyes and not a fit.

**Figure 7 materials-14-00005-f007:**
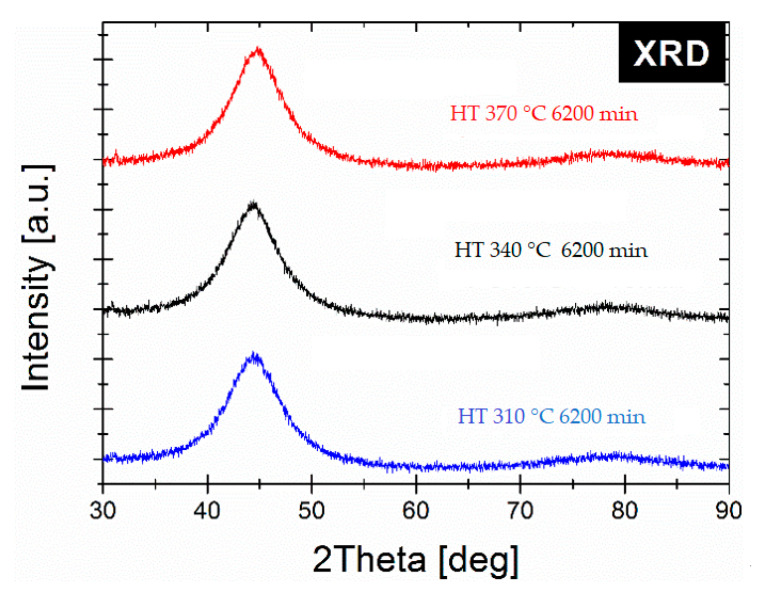
XRD patterns for relaxed glasses after 6200 min.

**Table 1 materials-14-00005-t001:** Comparison the magnetic properties of various FeNi-based alloys with obtained results.

Alloy	Ta [°C]/time	Ps [W/kg]	Bs [T]	Hc [A/m]	µ’	f_cut-off_ [kHz]	Ref
Fe_72_Ni_8_Nb_4_Si_2_B_14_	370/20 min	P_10/50_ = 0.092	1.09	3.95	3100	507	This work
Fe_72_Ni_8_Nb_4_Si_2_B_14_	440/20 min	P_10/50_ = 2.6	1.29	125	-	-	This work
Fe_70_Ni_10_Nb_4_Si_2_B_14_	as-cast	-	~1.62	-	-	-	[13]
Fe_60_Ni_20_Nb_4_Si_2_B_14_	as-cast	-	~1.44	-	-	-	[13]
Fe_56_Ni_24_Nb_4_Si_2_B_14_	440/60 min	P_1060_ = 0.12	~1.1	7	4000	-	[14]
Fe_56_Ni_24_Nb_4_Si_2_B_14_	440/60 min 200 MPa	-	1.3	-	16,000	-	[14]
Fe_77.4_Ni_8.6_B_14_	RA* 490/0.5s	-	1.7	2.6	-	-	[21]
Fe_68.8_Ni_17.2_B_14_	RA* 510/0.5s	-	1.54	4.4	-	-	[21]
Fe_60.2_Ni_25.8_B_14_	RA* 510/0.5s	-	1.37	3.2	-	-	[21]

* RA—Rapid Annealing with heating rate of 10^4^ K/s.

## Data Availability

The data presented in this study are available on request from the corresponding authors.

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
