# Peer review of "The Structure and Magnetic Properties of Rapidly Quenched Fe72Ni8Nb4Si2B14 Alloy"

_materials, 2020, doi:10.3390/ma14010005_

Round 1
Reviewer 1 Report
The manuscriptdeals with structural and magnetic characterization of amorphous (and also at the early stage of crystallization) FeNiNbSiB alloy.
They show that proper annealing below the crystallization temperature leads to the decrease of coercivity and core losses and cutoff frequency could increase with the annealing time.
The results are new and interesting and deserve publication in Materials.
However, there are some points that should be corrected prior to publication:
1.Numbers in figures should be increased. They are difficult to read when printed.
2.line 103: correct:"... dependence of coercivity on annealing temperature..."
3.line 123-124: last sentence is difficult to understand. Is it correct?
4.fig.2: The lines are the fit? If yes, the fit should be defined. If not, include: "the lines are only guide for eyes and not a fit".
5. line 136:"...for sample annealed at 370°C"
6. line 145: replace "420°C" with "440°C"
7. line 157: "based on the fact..."
8. line 175: "... dependencies on annealing..."
Author Response
Thank you very much for all suggestions. All points have been taken into account in the corrected version of the manuscript.
Reviewer 2 Report
In this work, the authors mainly studied the structure and magnetic properties of rapidly quenched alloys, XRD, DSC and TEM techniques are applied for analyzing corresponding results. Although some results are presented, it looks like an experimental report rather than a technical paper. The scientific contribution to the related field is limited, as the theoretical explanations are quite few. Additionally, the authors did not introduce the background, manufacturing process and applications of the amorphous alloy. Only a couple of parameters/temperatures were selected for experimental studies, why did you employ the parameters for heat treatment? It is better to use design of experiment method for detailed optimization analysis. The authors also did give solid analysis for TEM results, with only a brief description. Most of the references are relatively old, as many related publications can be easily find. Therefore, this is more likely an engineering report or short communication, rather than a scientific paper.
Author Response
Thank you very much for your comments. By taking into accounts comments of all reviewers the scientific contributions and explanations have been extended in the current state. Based on the over 30 years of experience in production various soft magnetic properties the heat treatment process and its parameters for defined chemical compositions has to be done usually iteratively up to reached the proper extremum of the magnetic properties value. In this paper we would like to show and focus mainly on application aspects and parameters that are measured. That was the reason why some of brief descriptions are present in the manuscript. A much more detailed structure studies including in situ TEM analysis and real space analysis of the high energy X-ray diffraction data together with APT and MOKE results will be published soon.
Reviewer 3 Report
The authors report the structural and magnetic properties of rapidly quenched Cu-free FeNi based Fe72-Ni8-Nb4-Si2-B14 soft-magnetic alloy. A systematic study of magnetic properties, namely, magnetic saturation, coercivity and core losses is conducted to determine the optimal heat treatment conditions for the reported alloy. Minimum coercivity and core losses are observed in the glassy state. The glass relaxation process is attempted to be understood by long-term annealing experiments as well. The manuscript is carefully written and the results are well presented, however, the reported combination of magnetic properties are not attractive from an application point of view. The following are my comments that could be addressed to further improve the quality of the paper:
* Line 173: please abbreviate APT and MOKE; include more details about how these tools could assist to understand the glass relaxation process with suitable references.
* Please discuss more on glass relaxation, reorganization of the magnetic domains, how trend in cut-off freq could be related to glass relaxation (Line 164, 167 and 168) and its significance from an application viewpoint, with suitable references.
* The XRD of the long-term annealed sample (mentioned in fig. 5) could be included too; or added as a supplement.
* Please tabulate the magnetic properties obtained from various heat treatment conditions, along with other reported values of FeNi based soft-magnetic alloys.
Further minor comments:
* Materials and methods: more details on the melting process and the purity of the constituent elements used for melting could be added.
* Line 36: mu-e is introduced without description.
* Line 46: ... caused by 'ac' high homogeneous ... (typographical error).
* Line 80: is this results and discussions ...
* Fig 4: index the XRD peaks (label phases).
* Fig 4: TEM- the 200 ring is not visible; kindly add circular markers to aid visualization. Also, indicate and scale the Fe-crystals (magnify if required) in the Figure (4, c).
* Line 177, is this the conclusion ...
* Line 180: Bs indicated twice (it is Bs and Ps ...)
Author Response
* Line 173: please abbreviate APT and MOKE; include more details about how these tools could assist to understand the glass relaxation process with suitable references.
* Please discuss more on glass relaxation, reorganization of the magnetic domains, how trend in cut-off freq could be related to glass relaxation (Line 164, 167 and 168) and its significance from an application viewpoint, with suitable references.
* The XRD of the long-term annealed sample (mentioned in fig. 5) could be included too; or added as a supplement.
* Please tabulate the magnetic properties obtained from various heat treatment conditions, along with other reported values of FeNi based soft-magnetic alloys.
Thank you very much for your comments. All the suggestions have been taken into account and added into the corrected version of the manuscript.
Further minor comments:
* Materials and methods: more details on the melting process and the purity of the constituent elements used for melting could be added.
* Line 36: mu-e is introduced without description.
* Line 46: ... caused by 'ac' high homogeneous ... (typographical error).
* Line 80: is this results and discussions ...
* Fig 4: index the XRD peaks (label phases).
* Fig 4: TEM- the 200 ring is not visible; kindly add circular markers to aid visualization. Also, indicate and scale the Fe-crystals (magnify if required) in the Figure (4, c). There are only several spots on the 200 ring, and the circular marker has been added. We suggest not to add the Fe-crystals scales, it comes to be less visible at the current figure size.
* Line 177, is this the conclusion ...
* Line 180: Bs indicated twice (it is Bs and Ps ...)
All the minors have been also corrected.
Round 2
Reviewer 2 Report
can be accepted